# A convergent evolutionary pathway attenuating cellulose production drives enhanced virulence of some bacteria

Nguyen Thi Khanh Nhu [1,2,3,16], M. Arifur Rahman[3,4,11,16], Kelvin G. K. Goh [5,6,16], Seung Jae Kim [7,8,16], Minh-Duy Phan [1,2,3], Kate M. Peters[1,2,3], Laura Alvarez-Fraga[2,3,12], Steven J. Hancock[2,3,13], Chitra Ravi[1,2,3], Timothy J. Kidd[2,3,14], Matthew J. Sullivan [5,6,15], Katharine M. Irvine [3,4], Scott A. Beatson [2,3], Matthew J. Sweet [1,3], Adam D. Irwin [3,9,10], Jana Vukovic [7,8] ✉, Glen C. Ulett [5,6] ✉, Sumaira Z. Hasnain [3,4] ✉ & Mark A. Schembri [1,2,3] ✉

Bacteria adapt to selective pressure in their immediate environment in multiple ways. One mechanism involves the acquisition of independent mutations that disable or modify a key pathway, providing a signature of adaptation via convergent evolution. Extra-intestinal pathogenic *Escherichia coli* (ExPEC) belonging to sequence type 95 (ST95) represent a global clone frequently associated with severe human infections including acute pyelonephritis, sepsis, and neonatal meningitis. Here, we analysed a publicly available dataset of 613 ST95 genomes and identified a series of loss-of-function mutations that disrupt cellulose production or its modification in 55.3% of strains. We show the inability to produce cellulose significantly enhances ST95 invasive infection in a rat model of neonatal meningitis, leading to the disruption of intestinal barrier integrity in newborn pups and enhanced dissemination to the liver, spleen and brain. Consistent with these observations, disruption of cellulose production in ST95 augmented innate immune signalling and tissue neutrophil infiltration in a mouse model of urinary tract infection. Mutations that disrupt cellulose production were also identified in other virulent ExPEC STs, *Shigella* and *Salmonella*, suggesting a correlative association with many *Enterobacteriaceae* that cause severe human infection. Together, our findings provide an explanation for the emergence of hypervirulent *Enterobacteriaceae* clones.

ExPEC is a leading human pathogen responsible for ~80% of all urinary tract infections (UTIs)[1] and ~25% of all bloodstream infections[2]. ExPEC are also the primary cause of meningitis in preterm neonates and the second most common cause of neonatal meningitis[3]. These infections are often associated with severe outcomes; pyelonephritis (kidney infection) is linked to high morbidity, the 30-day all-cause mortality rate of ExPEC bloodstream infection is ~16%[4], and neonatal meningitis has a mortality rate of 10–15% and neurological sequelae in 30–50% of cases in developed countries[5]. ExPEC are also associated with increasing antibiotic

A full list of affiliations appears at the end of the paper. ✉e-mail: j.vukovic@uq.edu.au; g.ulett@griffith.edu.au; sumaira.hasnain@mater.uq.edu.au; m.schembri@uq.edu.au

resistance[6], complicating treatment and increasing our dependence on last-line antibiotics.

ExPEC comprise a diverse array of strains from two major phylogroups (B2 and D) within the *E. coli* species[7]. Despite this, the majority of ExPEC strains that cause human infection belong to a small number of globally disseminated clones that can be differentiated based on their multi-locus sequence type (ST), including ST69, ST73, ST95 and ST131[8–11]. ST95 is the second most dominant global ExPEC clone, behind ST131[12–14], and comprises strains that cause both human and animal infections[15,16]. ST95 isolates generally possess fewer acquired antibiotic resistance genes compared to other global clones such as ST131, suggesting their virulence gene composition plays a key role in colonisation, transmission, and pathogenesis. Several virulent serotypes have been defined within ST95, including O1:K1:H7, O18:K1:H7 and O45:K1:H7[16,17]. Strains within these serotypes produce a K1 capsule comprised of polysialic acid and contribute to a significant burden of severe disease, including acute pyelonephritis, sepsis, and neonatal meningitis[18–20].

Isolates within the ST95 clone possess a mosaic composition of virulence factors, the majority of which are encoded on horizontally-acquired genomic islands (GIs). These include genes encoding various types of chaperone-usher fimbriae (e.g. type 1, P, S and F1C fimbriae), toxins (e.g. hemolysin and cytotoxic necrotising factor 1) and iron acquisition systems (e.g. siderophores and haem-binding proteins). ST95 also produce biofilms that enhance persistence during infection, a feature impacted by the production of an extracellular matrix (ECM) comprising curli amyloid fibres and/or the polysaccharide cellulose, which is modified by the addition of phosphoethanolamine (pEtN)[21]. The complexity of the ECM can vary; some ST95 such as the reference cystitis strain UTI89 produce an ECM composed of both curli and cellulose[22] while other ST95 strains such as MS7163 produce curli but not cellulose[23]. Intriguingly, the production of curli but not cellulose is linked to increased severity of disease, exemplified by strains MS7163 and S88 from the virulent ST95 O45:K1:H7 serotype that causes one third of all neonatal meningitis cases in France[19] and the highly virulent *E. coli* O104:H4 outbreak strain that infected nearly 4,000 people and caused 54 deaths in Germany in 2011[24]. Despite this, the impact of cellulose disruption on *E. coli* pathogenesis and the prevalence of this phenotype remains to be elucidated.

Here we used a curated collection of 613 publicly available genomes to generate a comprehensive recombination-free ST95 phylogeny, revealing eight distinct clades and a conserved K1 capsule genotype. Strikingly, we discovered a strong signature of disruption in cellulose production and pEtN modification via convergent evolution, and demonstrated how this phenotype drives enhanced virulence in two animal infection models, a rat model of neonatal meningitis and a mouse model of UTI. We further reveal that loss-of-function mutations in genes required for cellulose production are common in other virulent global *E. coli* clones, as well as in highly invasive *Shigella* and *Salmonella* spp., thus linking this pathoadaptive phenotype to augmented virulence.

## Results

### Phylogenomic analysis reveals ST95 comprises four major clades
We generated a curated dataset of 613 publicly available ST95 genomes (596 draft and 17 complete genomes) and used this to investigate clonal phylogeny employing read mapping and variant calling against the reference ST95 strain MS7163. This defined 29,057 core single nucleotide polymorphisms (SNPs), of which 8,450 SNPs (29.08%) were identified in recombination regions and removed for subsequent phylogenetic reconstruction. In total, 79 recombination regions were identified, accounting for ~22.6% of a typical ST95 genome, and primarily comprising mobile genetic elements such as prophages and GIs

(Supplementary Fig. 1). A final phylogenetic tree constructed with 20,607 recombination-free SNPs showed that the ST95 lineage is divided into eight well-supported clades, which we named 1 to 8 (Fig. 1a, left). Strains from clade 8 were most abundant and comprised 37.8% (232 strains) of the ST95 isolates. Together with clade 6 (29%; 178 strains), clade 3 (13.5%; 83 strains) and clade 4 (13.2%; 81 strains), the strains from these four major ST95 clades accounted for 93.6% (574/613) of genomes in the database, likely reflecting their increased association with disease pathogenesis. This phylogeny and clade structure was largely congruent with alternative analyses using fastbaps[25] and popPUNK[26], with discrepancies likely associated with the lower resolution of these alternative methods (Supplementary Fig. 2).

We determined a core genome of 3430 protein-coding sequences (CDS) shared by >99% of strains and a pangenome of 24,049 unique CDS from the 612 ST95 strains. A gene accumulation curve showed the ST95 pangenome is open (Supplementary Fig. 3), inferring additional genes would be detected if more genomes were examined. We also examined the source of isolation for the ST95 strains in the context of our phylogeny. No geographic clustering was observed between the major clades. All clade 1 strains and 75% of clade 2 strains were from non-human sources (Supplementary Data 1). In contrast, clades 3–8 contained strains from human and non-human (predominantly avian and environmental) sources, with most strains from these sources forming separate well-supported branches. In silico capsule typing revealed that all strains possess a K1 capsule, with no evidence of recombination within this region (Supplementary Fig. 1). In contrast, ST95 contains a diverse range of O and H serotypes, with genes encoding these structures located in recombination regions (Supplementary Fig. 1). Most ST95 serotypes clustered within a single clade or formed a well-supported subclade within a larger clade (Fig. 1a). In addition, ST95 harbour a wide range of ExPEC-associated virulence factors, the majority of which are located within GIs (Supplementary Fig. 4, Supplementary Data 2).

### Convergent evolution drives loss-of-function in cellulose production genes in ST95
We previously showed that the genome sequenced O45:K1:H7 strains MS7163 and S88 possess a frameshift mutation in the cellulose synthase gene *bcsA* leading to disruption of cellulose production[23]. Strikingly, our investigation here revealed that this mutation (at amino acid 186 in BcsA) is present in 218/232 (93.7%) strains from clade 8 (including all other O45:K1:H7 strains) (Fig. 1a). We extended this analysis by examining the coding sequence of cellulose synthesis, secretion and pEtN modification genes within the divergent *bcsRQABZC-bcsEFG* operons[21,27,28]. In total, there were 27 independent loss-of-function mutations that occurred in 339/613 (55.3%) of ST95 strains (Fig. 1b). In addition to the BcsA frameshift 186 mutation, other high-frequency loss-of-function mutations include a BscA W55* mutation (present in 11 clade 3 strains), a frameshift mutation at amino acid 418 in the BcsG pEtN transferase cellulose modification enzyme (present in 26 O1:H7 strains from clade 6), and a Q178* mutation in the cellulose biosynthesis protein BcsQ (present in 24 strains from clade 3 and clade 4) (Fig. 1a, b). Taken together, identification of these loss-of-function mutations reveals a signature of convergent evolution towards ablation of cellulose production and pEtN modification in ST95.

As the attenuation of cellulose production has been observed in some pathogenic *E. coli* that cause severe disease[23,24], we probed our genomic dataset by comparing the integrity of this locus in human ST95 isolates sourced from blood, urine and faeces, of which there were 333 isolates with appropriate metadata. Loss-of-function mutations that disrupt cellulose production or its modification were significantly associated with isolates sourced from blood (125/232)

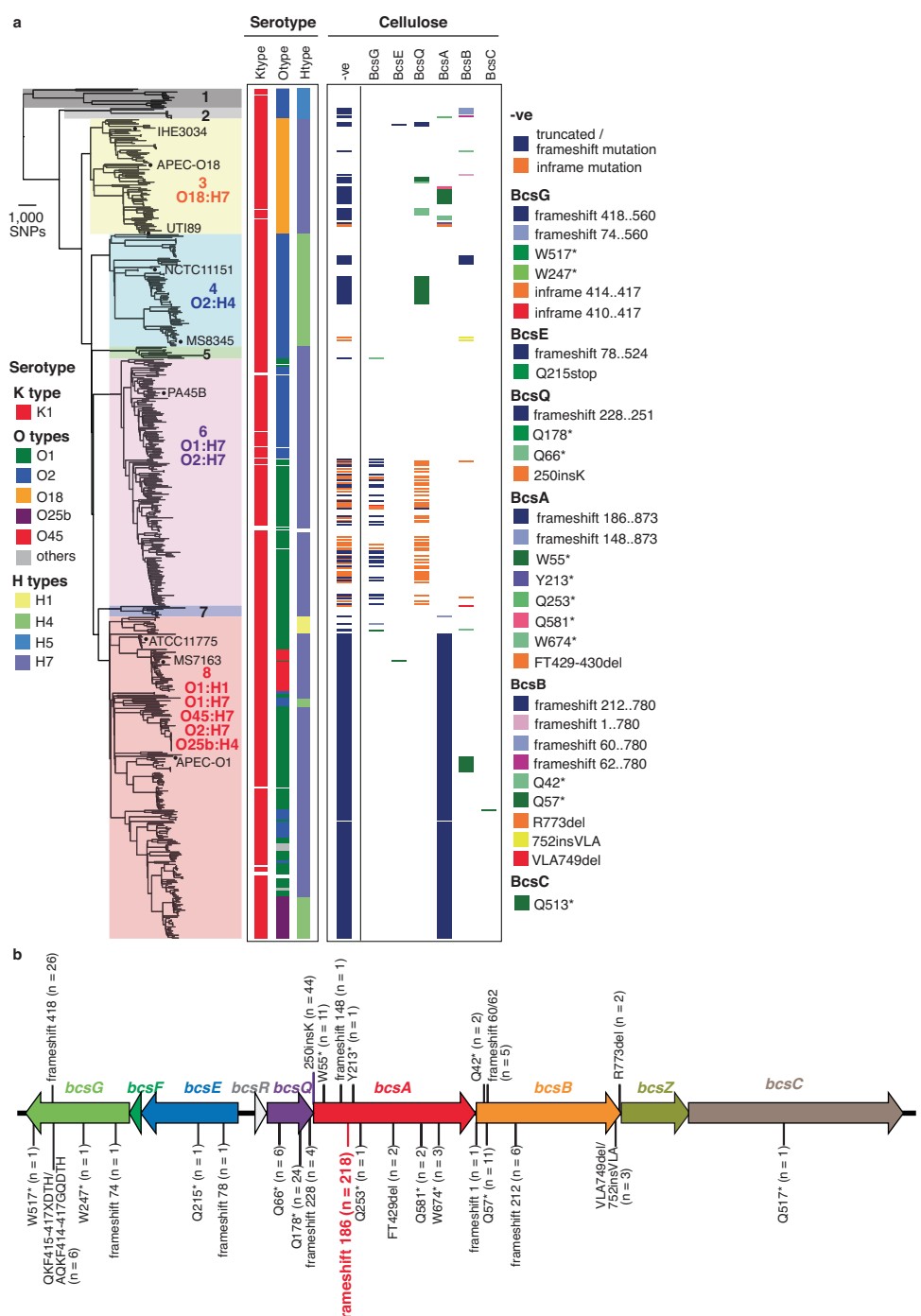

**Fig. 1 | Phylogenetic tree of ST95. a** Maximum likelihood tree of 612 ST95 strains constructed using 20,607 recombination-free core SNPs with 1000 bootstrap and rooted using ED1a as an outgroup. The eight ST95 clades are coloured for clarification. Overall, the mean pairwise nucleotide divergence between genomes in the complete dataset was 1.5% (range 0–3.7%), and between genomes within each clade was 0.9% (range 0.52–1.67%). Clades 1 and 2 exhibited the greatest divergence, with a mean pairwise nucleotide divergence to genomes from other clades of 3.09% and 2.37%, respectively. The remaining clades were more closely related, with a mean nucleotide divergence of 1.6% (range 1.47–1.7%). The serotype of each strain is indicated with respect to capsule (K), O antigen (O) and flagellar (H), with the major serotype/s noted. The mapping of independent mutations in genes encoding proteins required for cellulose biosynthesis (BcsQABC) and modification (BcsEG) is also shown. A summary of the predicted phenotype of cellulose attenuation (-ve) is indicated. **b** Truncated mutations in cellulose biosynthesis and pEtN modification genes in ST95.

compared to urine (28/78) and faeces (7/23) (Chi-square test, 2-tailed *P*-value = 0.0049), thereby supporting a hypothesis of increased virulence (Supplementary Fig. 5a). In contrast, we did not find a significant association of cellulose gene attenuation in ST95 isolates sourced from animals (30/47) or the environment (11/22) in our dataset (Chi-square test, 2-tailed *P*-value = 0.3) (Supplementary Fig. 5b).

## Cellulose disruption enhances ST95 virulence in a rat model of neonatal meningitis

To experimentally investigate the impact of cellulose disruption on ST95 virulence, we began by correcting the most common loss-of-function mutation, a G deletion at nucleotide 556 that creates a premature stop codon at amino acid 186 in the BcsA synthase, in the clade

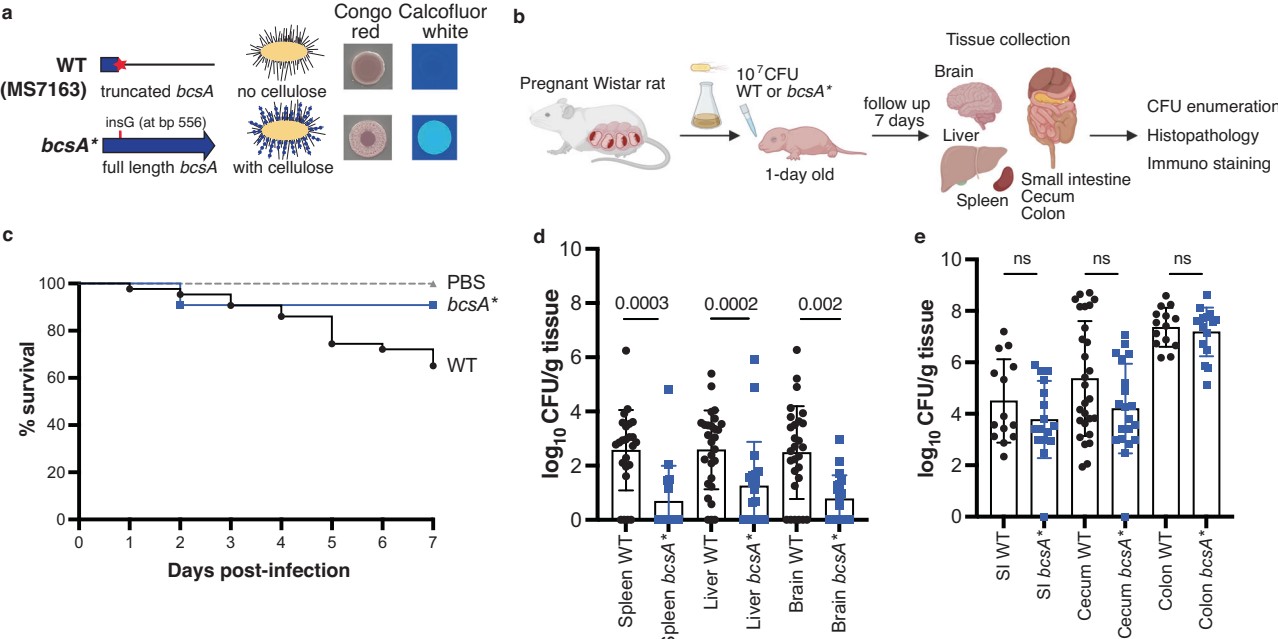

**Fig. 2 | Cellulose disruption enhances ST95 virulence. a** Genotypes and phenotypes of wild type MS7163 (WT) and its cellulose corrected mutant *bcsA**. **b** Diagram of a neonatal rat meningitis model (created with BioRender). One-day old Wistar rat pups were infected with 10⁷ CFU bacteria and followed for 7 days for survival. At day 7 post infection, the pups were euthanized to collect the brain, liver, spleen, small intestine (SI), caecum and colon, from which bacteria were enumerated.

**c** Percentage of survival of pups during the 7 day infection period (WT, 43 pups; *bcsA**, 22 pups; PBS, 12 pups). **d** Enumeration of WT or *bcsA** loads in the spleen, liver and brain. **e** Enumeration of WT or *bcsA** loads in the gastrointestinal tract. Data are presented as mean ± SD with each point representing a single pup (1–3 litters, with 10–16 pups/litter). Error bars *P*-values were determined with two-tailed Mann–Whitney test. Source data are provided as a Source Data file.

8 strain MS7163. This wild-type strain MS7163 (referred to as WT hereafter) did not produce any cellulose. The resulting strain, designated *bcsA**, produced cellulose and exhibited a cellulose-positive phenotype characterised by its red, dry and rough colony morphotype on yeast extract and casamino acid (YESCA) agar containing the dye Congo Red and its fluorescence under UV light when stained with calcofluor white (Fig. 2a).

We next examined the role of cellulose in ST95 using a neonatal rat model of dissemination and meningitis. One-day old Wistar rat pups were orally infected with 10⁷ CFU of either WT or *bcsA** strains and followed for 7 days (Fig. 2b). The survival rate at day seven post infection was significantly lower for pups infected with WT (65.1%) compared to pups infected with *bcsA** (90.9%; *P* = 0.0375, Mantel-Cox test); all pups survived in the PBS control group (Fig. 2c). Examination of bacterial dissemination to different tissues revealed WT was more invasive; more pups were recovered with WT in the spleen (20/24, 83.3%), liver (25/28, 89.3%) and brain (22/28, 78.6%) compared to pups infected with *bcsA** (spleen: 5/16 [31.2%]; liver: 12/20 [60%]; brain: 11/20 [55%]) (Fig. 2d). Bacterial loads from pups infected with WT were also significantly higher in these tissues compared to pups infected with *bcsA** (*P* = 0.0002 [spleen], *P* = 0.002 [liver], *P* = 0.0003 [brain]; Fig. 2d). Despite the differences in bacterial loads in the spleen, liver and brain, the WT and *bcsA** strains colonised at equivalent levels in the small intestine, caecum and colon (Fig. 2e). As the presence or absence of cellulose may impact K1 capsule production, we also tested K1 capsule production by ELISA using a specific antibody[29]. In these experiments, both WT and *bcsA** produced equivalent amounts of K1 capsule (Supplementary Fig. 6). Finally, as a control to demonstrate the impact of the K1 capsule on virulence in our model, we disrupted K1 capsule production by mutating the *kpsD* gene in MS7163 (effectively creating a double capsule and cellulose mutant). This mutant was completely attenuated in our model (Supplementary Fig. 7). Together, these results suggest that the differences in WT versus *bcsA**

dissemination were not caused by differences in the load of intestinal infection nor the capacity to produce the K1 capsule.

## Cellulose disruption impairs gut mucosal barrier integrity

We next examined the mechanism driving increased dissemination from the intestine by WT compared to *bcsA**. Although bacterial loads were similar in the intestine, a significant increase in the weight of the small intestine following infection with the WT strain compared with *bcsA** suggested an increase in pathology (Fig. 3a). The increase in pathology was confirmed through histology; infection with WT resulted in widespread small intestinal and colonic damage with prominent mucosal erosion, villus destruction, and increased inflammatory infiltration (Fig. 3b; Supplementary Fig. 8a). Although some inflammatory infiltration was apparent, this was not observed to the same extent in the *bcsA**-infected animals compared to WT. The destruction in crypt structure was associated with increased apoptosis; this was more apparent in the small intestine, demonstrated using TUNEL staining in the WT-infected animals, but absent in *bcsA**-infected animals (Fig. 3c, Supplementary Fig. 8b). There was also extensive loss in tissue architecture in the colon (Supplementary Fig. 8c). Modest changes in Iba1 staining (macrophages) were observed in both small intestine and the colon between WT and *bcsA**-infected animals, with some variability between animals (Fig. 3d, Supplementary Fig. 8c). We noted disrupted localisation of the integral tight junction protein E-cadherin, which was most apparent in the small intestine (Fig. 3d, Supplementary Fig. 8c). These changes were accompanied by a slight decrease in goblet cells in the small intestine (Fig. 3e) and colon (Supplementary Fig. 8d) of WT-infected animals. Flow cytometry analyses highlighted an increase in B cells and CD4 + T cells, particularly inflammatory monocytes in the colon (Supplementary Fig. 9). The increase in inflammation and pathology is consistent with disruption of the epithelial paracellular barrier. Taken together, the observations indicate significant inflammation in the intestine, which is associated

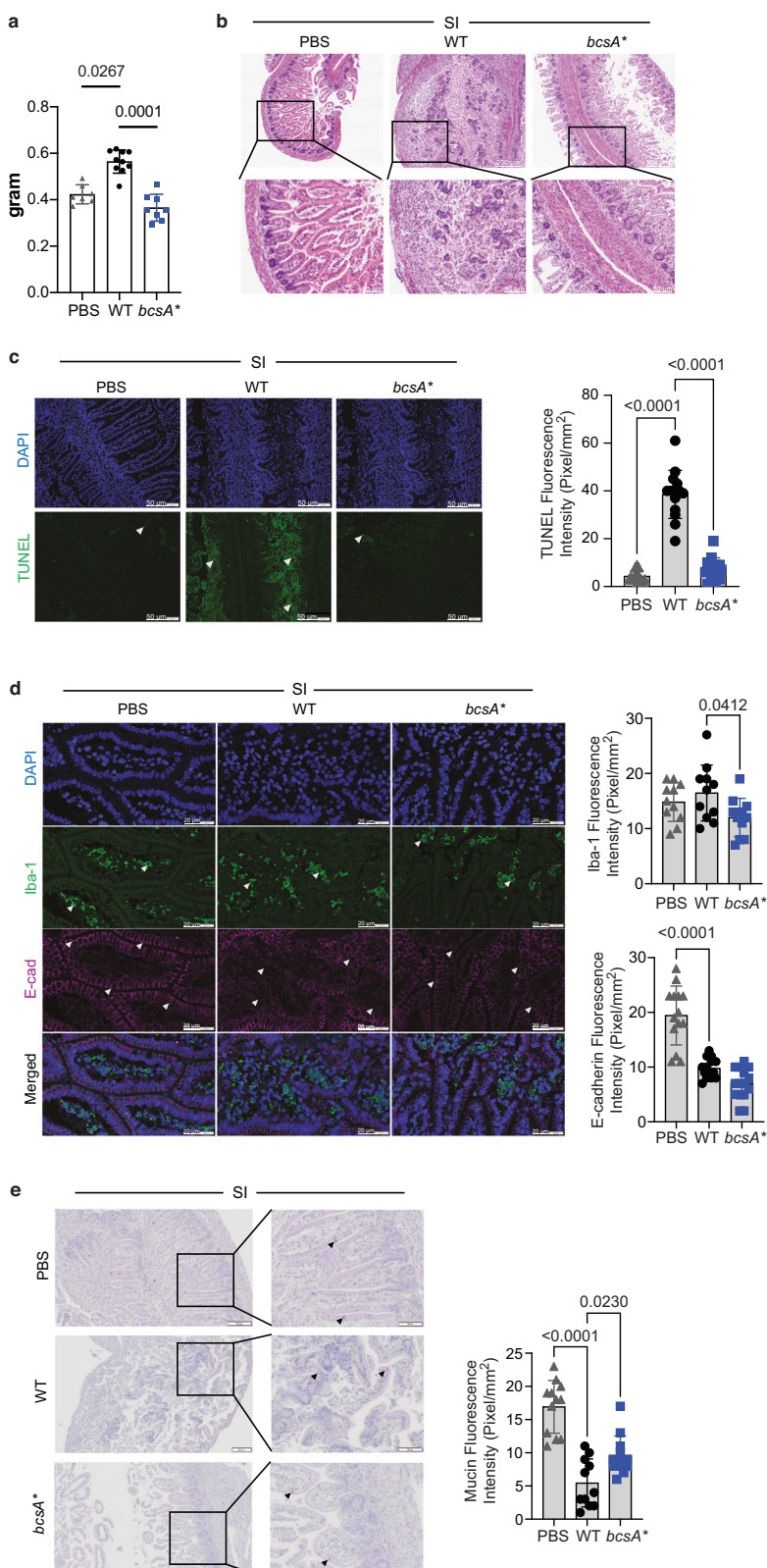

**Fig. 3 | Cellulose disruption impairs the barrier integrity in the rat pup small intestine. a** Weight of small intestines harvested from rat pups 7 days post-infection ($n = 7$–11). **b** Representative tissue pathology (H&E) micrographs (scale bars 200 μm in top panel and 80 μm in bottom panel). **c** apoptosis (TUNEL) staining images (white arrowheads indicating cell death, scale bars- 50 μm) and relative quantification of TUNEL fluorescence intensity (Pixel/mm$^2$). **d** Immune-fluorescent staining images (scale bars 20 μm) showing macrophage (white arrowheads) infiltration and tight junction protein, E-cadherin expression (white arrowheads). **e** Mucus secreting goblet cells (black arrowheads) staining (Alcian Blue-Periodic Acid Schiffs) and relative quantification of Mucin fluorescence intensity of the small intestines of 7 days old rat pups infected with PBS, WT or *bcsA\**. Data are presented as mean ± SD (one-way ANOVA, two-three independent experiments, $n = 4$–10). Source data are provided as a Source Data file.

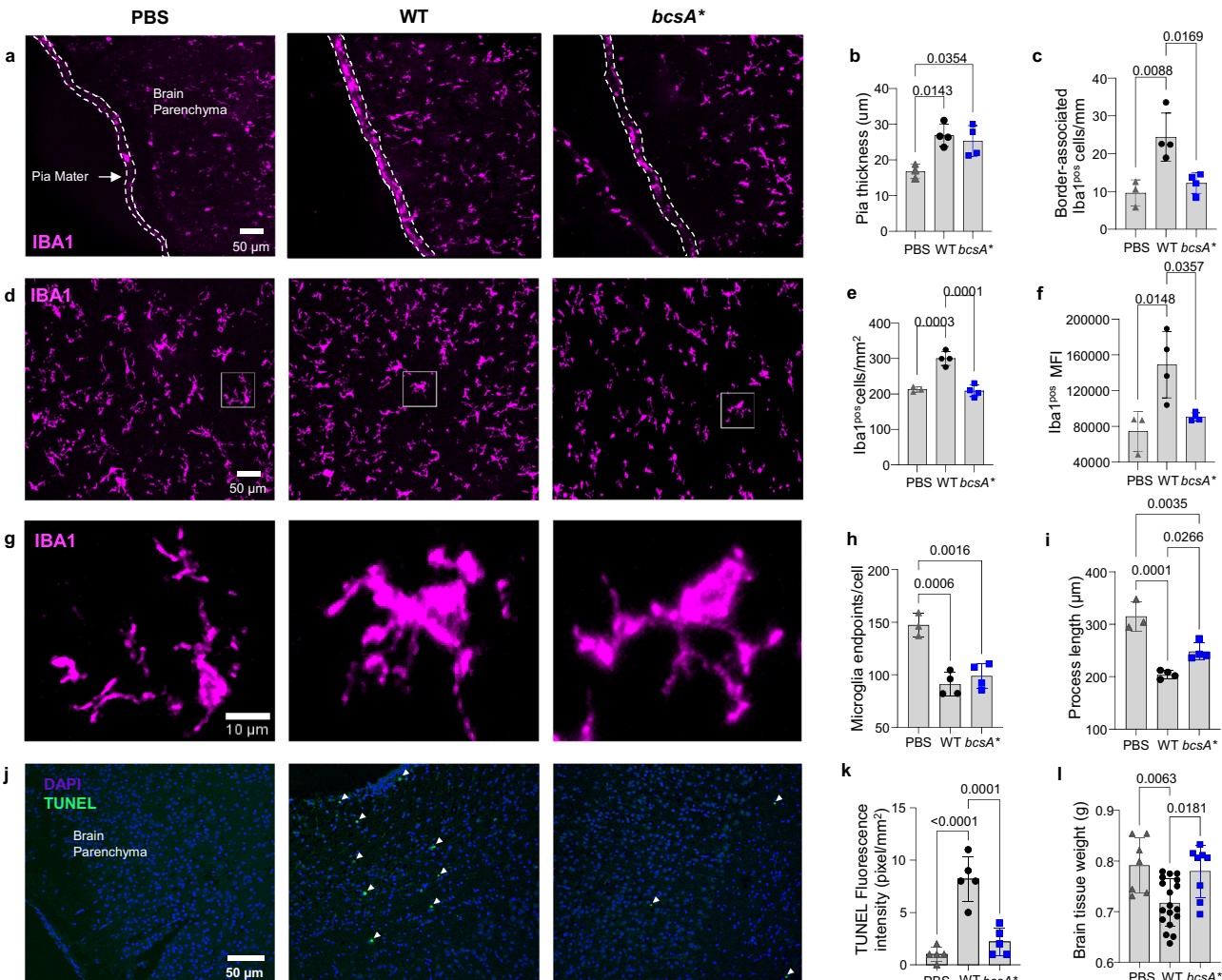

**Fig. 4 | Cellulose disruption leads to increased neuroinflammation in the rat pup brain shown by enhanced meningeal and parenchymal Iba1^pos cell counts, increased microglia activity, and altered microglia morphology into its pro-inflammatory state. a** Representative immunofluorescence images, quantifications of pia mater thickness (**b**) and border-associated Iba1^pos macrophage counts (**c**). **d** Parenchymal Iba1^pos microglia in the cortex, quantification of cell number (**e**) and mean fluorescence intensity (MFI) (**f**). **g** Microglia morphology analyzed by quantifying the average number of endpoints per cell (**h**) and process lengths in the cortex (**i**). **j** Apoptosis (TUNEL) staining images and (**k**) their TUNEL fluorescence intensity. Each datapoint (**b**, **c**, **e**, **f**, **h**, **i** and **k**) is presented as the mean of three separate sections from individual pups ($n = 3–5$). **l** Weight of rat pup brains harvested 7 days post-infection ($n = 7–18$). For all graphs, data are presented as mean ± SD and were analysed using a one-way ANOVA with Bonferroni's correction. Individual $p$-values are displayed where significance is present. Source data are provided as a Source Data file.

with impaired intestinal barrier function and infection. Notably, this was accentuated in the animals infected with the WT strain compared to $bscA^*$.

### Increased intestinal permeability is associated with augmented systemic infection

We next examined the livers of WT and $bcsA^*$-infected animals as an indicator of barrier defence breakdown. Consistent with the notion that impaired paracellular permeability would lead to increased WT translocation and systemic infection, we observed an increase in inflammatory cell infiltration with H&E staining in the liver of both WT and $bcsA^*$-infected groups (Supplementary Fig. 10a). Moreover, there was an increase in cell death in the liver of animals infected with WT compared to $bcsA^*$ (Supplementary Fig. 10b). CD11b/c^hi neutrophils and CD172a^hi monocytes and macrophages were modestly increased in the livers of WT-infected animals, with a significant increase in HIS48+ classical monocytes in WT compared to $bcsA^*$ and PBS control groups. Macrophages (defined by CD4 expression[30,31]) were the only MHCII+ cell in the liver, and MHCII expression was strongly upregulated in

infected animals. CD4 + T cells and CD45R + B cells were also reduced in WT-infected rat pups (Supplementary Fig. 10c).

We also evaluated the impact of infection of the central nervous system (CNS) by examining the degree of (neuro)inflammation in the brains of pups colonised with WT or $bcsA^*$. Mirroring our observations in the intestines and liver, WT-infected brains exhibited profound inflammatory changes, which were markedly attenuated in $bcsA^*$-infected animals. In particular, the brains of WT-infected pups displayed (i) prominent meningeal thickening and a significantly greater number of border-associated macrophages (Fig. 4a–c), (ii) significant expansion and activation of microglia, resident phagocytes of the brain (Fig. 4d–f), and (iii) a shift towards amoeboid microglial morphology (Fig. 4g–i). Importantly, the neuroinflammatory changes in pups infected with WT were found to have a significant negative impact on postnatal brain development, with increased neural cell death and reduced overall brain weights (Fig. 4j–l). In sharp contrast, infection with $bcsA^*$ resulted in only a very modest immune response, with brain weights remaining unaltered and similar to vehicle controls, indicative of normal gross brain development.

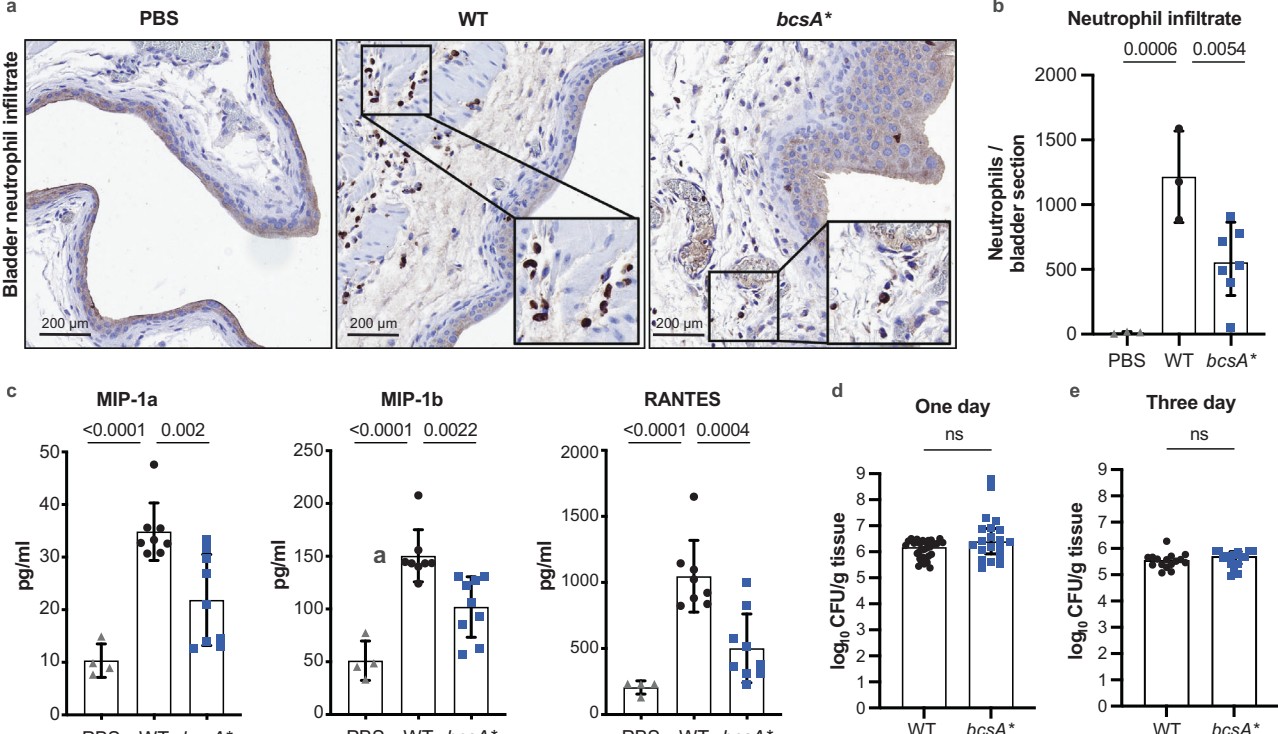

**Fig. 5 | Cellulose disruption does not affect colonisation but enhances neutrophil infiltration of the bladder in the murine model of UTI. a** Representative bladder tissue sections from mice that received PBS, WT or *bcsA**. Sections were stained with anti-Ly6G antibodies to show neutrophils (brown). **b** Neutrophil infiltrates were quantified for sections of whole bladder and presented as mean ± SD. Each point represents an average count for duplicate bladder sections from a single mouse (*n* = 3–7). Data were analysed using a one-way ANOVA with Dunnett multiple comparisons, with *p*-values displayed. **c**, Cytokine responses in the bladder (*n* = 4–9) in response to infection presented as mean ± SD. Bladders of mice infected with WT exhibited higher levels of chemotactic cytokines MIP-1a, MIP-1b

and RANTES compared to infection with *bcsA**. Data were analysed using a one-way ANOVA with Dunnett multiple comparisons, with *p*-values displayed. **d** Bacterial load recovered from the bladder of mice infected with WT and *bcsA** at 1-day post-infection. Data are pooled from two independent experiments (*n* = 20 for each strain). Presented as mean +/− SD, ns = not significant; compared using the two-tailed Mann–Whitney test. **e** Bacterial load recovered from the bladder of mice infected with WT and *bcsA** at 3-days post-infection (*n* = 16), presented as mean +/− SD, ns = not significant; compared using the two-tailed Mann–Whitney test. Source data are provided as a Source Data file.

## Cellulose attenuation is associated with augmented immune cell infiltration in the bladder

To extend our discovery linking cellulose disruption with enhanced severity of disease, we employed a mouse model of acute UTI to examine the impact of this phenotype on infection and immunopathology in the bladder. Analysis of bladder tissue at 1-day post-infection revealed significant neutrophil infiltration into the bladder in mice infected with WT, with this being attenuated in *bcsA**-infected animals (Fig. 5a, b). Neutrophil influx is a hallmark of local inflammation and thus we measured the levels of major inflammatory cytokines in the bladder tissue. Strikingly, mice infected with WT exhibited significantly higher levels of the chemotactic cytokines MIP-1a, MIP-1b and RANTES in bladder tissue compared to mice infected with *bcsA** (Fig. 5c). These significant differences in local tissue chemotactic cytokine production and cellular infiltrate in mice infected with WT compared to *bcsA** occurred despite the mouse bladders harbouring equivalent bacterial loads at 1-day post-infection (Fig. 5d) and 3-days post-infection (Fig. 5e) between these groups.

## Mutations disrupting cellulose production also occur in other virulent clones

We hypothesised that mutations disrupting cellulose production and pEtN modification would not be limited to ST95, but rather this genotype could be a signature of other virulent clones. Therefore, we compiled a database comprising 100 genomes randomly chosen from each of the 76 most common *E. coli* STs in Enterobase[32] and investigated the integrity of genes in the *bcsRQABZC-bcsEFG* cellulose

biosynthesis operons in these genomes by screening for loss-of-function mutations. Mutations that disrupt at least one gene involved in cellulose production or pEtN modification were identified in 97.3% (74/76) STs (Fig. 6a, Supplementary Data 3), with 18.4% (14/76) STs containing >80% of strains with cellulose-disrupting mutations (Fig. 6a). Further examination of the precise mutations in these strains revealed a correlation pattern shared within individual or closely related STs (Fig. 6a). For example, ST167, ST617 and ST744, all of which are in the ST10 complex (phylogroup A), contain the same nonsense mutations in BcsQ and BcsZ. All strains from ST300, ST453, ST655, ST1792 (phylogroup B1), ST1193 (phylogroup B2) (Fig. 6b), ST32 (phylogroup D) and ST11 (phylogroup E) have unique combinations of loss-of-function mutations within each ST. In addition, strains from ST69 and ST405 (phylogroup D), ST141 (phylogroup B2) and ST678 (phylogroup B1) possess cellulose nonsense mutations that can be mapped to defined branch points based on genomic relatedness (Fig. 6c–f). Taken together, these STs exhibit a phylogenetic clustering pattern consistent with a model describing clonal expansion following the acquisition of cellulose loss-of-function mutations.

We next performed a similar analysis on *Shigella* and *Salmonella*, two genera closely related to *E. coli* that also belong to the *Enterobacteriaceae* family. All *Shigella* spp. STs examined contain loss-of-function mutations in at least one of the nine cellulose genes (Supplementary Fig. 11, Supplementary Data 4). In *Salmonella*, mutations that disrupt one or more genes involved in cellulose production or pEtN modification were identified in 90% (90/100) STs, with 31% (28/90) STs containing >90% of strains with cellulose-disrupting mutations

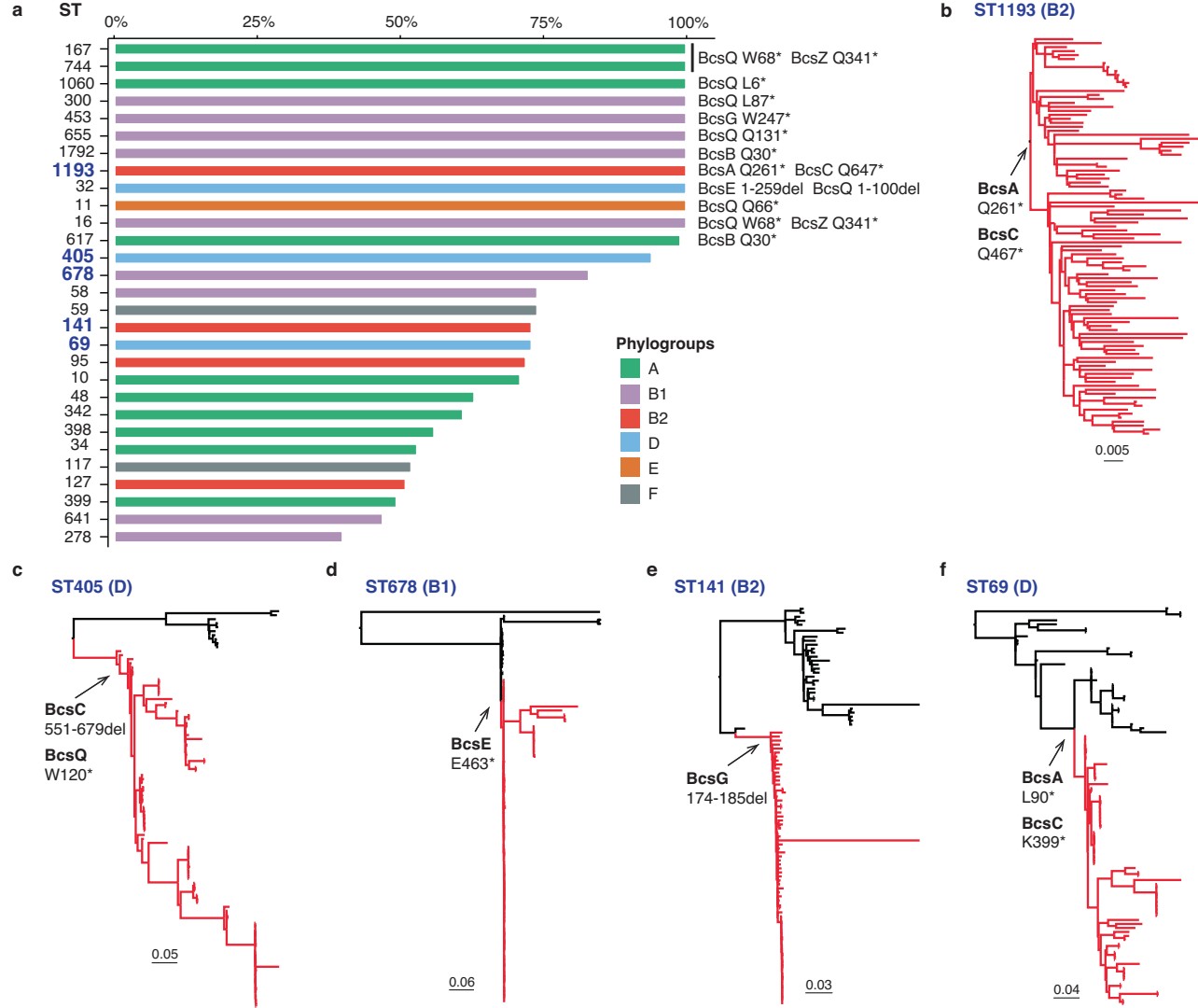

**Fig. 6 | Loss-of-function mutations in cellulose biosynthesis in *E. coli*. a** Percent of strains in the 76 most common STs represented in Enterobase that possess an attenuating mutation in at least one gene in the cellulose biosynthesis and pEtN modification operons (data shown for STs with >25% strains having disruptions).

**b**–**f** Examples of cellulose disruption mutations in the clonal expansion profile of pathogenic *E. coli* STs (cellulose mutations indicated as red branches). Source data are provided as a Source Data file.

(Supplementary Fig. 12, Supplementary Data 5). Thus, loss-of-function mutations in cellulose biosynthesis or pEtN modification are common in *Enterobacteriaceae* that cause invasive human infections.

## Discussion

Here we report on convergent evolution linking disabling of cellulose biosynthesis to enhanced bacterial virulence. By generating a comprehensive ST95 phylogeny, we identified K1 capsule production as a conserved ST95 feature. Furthermore, we show that loss of cellulose production and/or pEtN modification is an evolutionary driver in ST95 and other *E. coli* clones, as well as other common *Enterobacteriaceae* species that cause severe invasive disease.

ST95 comprises a narrow spectrum of serotypes based on O, H and K antigens. The conserved K1 capsule is an alpha-2,8-linked sialic acid, with the structure identical to human polysialic acid typically present on neural cell adhesion molecule, a glycan found on the surface of neurons, glial cells and natural killer cells[33]. This unique structure explains its poor immunogenic characteristics and its role in pathogenesis, with the K1 capsule contributing significantly to ExPEC infection of the blood and brain[34–38]. Compared to the capsule region

of other ExPEC clones such as ST131, which has been significantly recombined[13,39], the K1 capsule locus of ST95 strains is highly conserved and we found no evidence of recombination (Supplementary Fig. 1), a finding consistent with a recent study that examined a different ST95 strain collection[40]. In contrast, the genes encoding O antigen biosynthesis reside in a chromosomal recombination hotspot and their diversity reflects adaptation via host immune evasion[41–44]. Similarly, the contribution of diverse O antigens to pathogenesis depends on their type and structure, and the pairing of specific O-K types is frequently linked to ExPEC virulence[36,38,45–47]. For example, the O1:K1 serotype is one of the most common antigen combinations found in *E. coli* that cause acute pyelonephritis in children[20], while O18:K1 is associated with neonatal meningitis[36,48]. Taken together, the conservation of K1 capsule in ST95 together with its narrow range of O-H serotypes suggests these features are under selection pressure linked to a fitness advantage during infection.

Bacterial cellulose is a homopolysaccharide of β-1,4-linked glucosyl residues modified with a pEtN group[21]. The genes involved in cellulose production and pEtN modification are encoded on the *E. coli* chromosome in two divergent operons, *bcsRQABZC* and *bcsEFG*[49,50].

We hypothesise that the large diversity within our ST95 genome dataset likely accounts for the cellulose loss-of-function mutations occurring in just over half of the isolates we examined. For example, the dataset includes human, animal and environmental isolates derived from multiple sources, comprises isolates derived from 20 different countries, and spans a broad timeline of isolation. Indeed, it is likely that in some of these settings, cellulose production may be advantageous. Based on our data, we suggest one example may be asymptomatic colonisation of the intestinal tract.

The function of cellulose is linked to the production of curli amyloid fibres, with both components combining to form an extracellular matrix that enhances *E. coli* biofilm formation[51–53]. Further analysis of cellulose and curli has revealed they contribute differently to *E. coli* pathogenesis and innate immune induction. Curli are highly proinflammatory amyloid fibres[54–56], induce Toll-like receptor (TLR)1/TLR2 complex-mediated immune responses[57] as well as the formation of immunogenic complexes with DNA to stimulate autoimmune responses[58], and can neutralize human cathelicidin (LL-37), a soluble antimicrobial peptide that protects against UTI[55,59]. Curli also play a role in bladder colonisation in a murine infection model[23,53] and are associated with urosepsis and bloodstream infection[54,60]. In contrast, cellulose production can dampen curli-stimulated immune induction[55,61]. Here, we show that loss-of-function mutations in genes in cellulose production and pEtN modification have occurred via multiple independent events in the ST95 clone, generating a strong signal for convergent evolution and providing a mechanism to explain the enhanced virulence phenotype of ST95. By correcting the most common mutation in the cellulose synthase gene *bcsA* to restore cellulose production in the prototypical ST95 strain MS7163, we further demonstrated how this impacts virulence. We hypothesize that the attenuation of cellulose production may unmask and enhance the proinflammatory effect of curli (as well as other cell-surface factors). However, as curli regulation is complex and often temperature-regulated, this remains to be demonstrated experimentally.

Several studies have reported an increase in intestinal permeability with infection. Here, we show that while cellulose disruption does not affect colonisation of the intestine, it has a profound effect on the intestinal inflammation that follows. WT colonisation in the intestine resulted in cell death and widespread inflammation along with mucosal erosion. The destruction of the intestinal architecture accompanied by the increased inflammatory cell infiltration was largely absent in animals infected with the *bcsA\** cellulose producing strain. We have previously shown that inflammation can have a profound effect on the intestinal epithelial paracellular permeability[62–64]. The small intestine has higher permeability than the colon. The disruption in tight junctions, as well as the diminished goblet cells was more pronounced with WT infection in the small intestine and is indicative of a disrupted epithelial paracellular barrier. Overall, this explains the increased systemic infectious load we observed in WT compared to *bcsA\**-infected rat pups.

Systemic infection with *E. coli* K-12 in newborn rat pups has been shown to negatively impact the generation of new neurons in neonates, resulting in lower neuronal counts in the adult brain[65]. Here we demonstrate a direct link between increased gut permeability and impaired brain development in neonatal rats following oral ingestion of the virulent ExPEC ST95 strain MS7163. Similar to our observations in gut and liver, inflammation in the brain parenchyma and meninges was dampened by restoring the capacity to produce cellulose. Brains of neonates colonised with WT were hallmarked by profound inflammatory changes which had a significant negative impact on postnatal brain growth. In sharp contrast, neuroinflammatory changes in *bcsA\**-infected pups were markedly attenuated and these pups displayed normal brain weights, likely also linked to reduced infection loads. The postnatal period is critical for healthy brain development and brain weight/size during the first postnatal week correlates with the net number of neurons in adulthood. Together, our evidence demonstrates that alteration of the neural milieu by ExPEC in neonates is particularly augmented by the inability to produce cellulose. Future studies can now examine how loss of cellulose production by virulent ExPEC stifles postnatal brain development during meningitis and investigate how neuroinflammation, supressed neural cell proliferation and/or cell death contribute to adverse developmental outcomes in this pathology.

Although the restored synthesis of cellulose did not attenuate *bcsA\** for colonisation of the murine bladder, its production was associated with significantly reduced neutrophil influx and cytokine production. Cytokine expression and inflammatory cell infiltration into the bladder during UTI depend on the species of causal bacteria[66]; however, a central role for bacterial cellulose in shaping host innate immune responses has not been defined. Here, we demonstrate that the production of cellulose is a key modulator of host innate immune signalling during infection of the bladder.

Our genomic analyses showed that the cellulose-negative genotype is present in multiple *E. coli*, *Shigella* and *Salmonella* clones associated with severe invasive disease. In ST95, the frameshift mutation from amino acid 186 in BcsA was conserved in most clade 8 strains, suggesting it was acquired by the common ancestor of clade 8 and amplified following clonal expansion. We observed a similar profile of clonal expansion following loss-of-function mutations in cellulose production and pEtN modification in other ExPEC, including ST1193 and ST141 (phylogroup B2), and ST69 and ST405 (phylogroup D). ST1193 is a recently emerged fluoroquinolone resistant UPEC clone[67–70], and frequently isolated from neonates with invasive infections[71]. ST141 is a hybrid Shiga toxin-producing *E. coli* (STEC) and UPEC lineage associated with diarrhoea and UTI in humans[72]. ST69 is another common ST associated with bloodstream infections, but features limited antibiotic resistance and a reduced virulence gene repertoire compared to many other ExPEC B2 STs[11,73]. ST405 is an emergent UTI and urosepsis lineage associated with ESBL gene carriage including $bla_{CTX-M-15}$ and $bla_{NDM}$[10,74,75]. In *Shigella* spp., while *Shigella boydii* carries the same *bcs* operon as *E. coli*, three other species have frameshift mutations in *bcsA* (*Shigella flexneri* and *Shigella sonnei*) or other *bcs* genes (*Shigella dysenteriae*), which could reflect adaptation of these organisms to their intracellular lifestyle[76]. In *Salmonella*, several studies have shown that cellulose disruption promotes virulence[77–79]. Here, we extended these findings and demonstrate that cellulose disruption mutations occur in all human-restricted *S. enterica* serovars causing typhoid fever (*S.* Typhi) and paratyphoid fever (*S.* Paratyphi A, B and C). Interestingly, disruption of the BcsG pEtN transferase is associated with clonal replacement of *Salmonella* Typhimurium ST313 lineage 1 by lineage 2, which is a major cause of bloodstream infection in Africa[80,81]. Thus, loss-of-function in cellulose production is a pathoadaptive mechanism that shares a correlative association with many *Enterobacteriaceae* that cause severe human infection.

## Methods

### Ethics approvals
All work involving rats was approved by the University of Queensland Animal Ethics Committee (AEC approval number SCMB/127/19). All work involving mice was approved by the University of Queensland Animal Ethics Committee (AEC approval number SCMB/259/19) and the Griffith University Animal Ethics Committee (AEC approval number MSC/01/18).

### Genome data
Genome sequence data for 850 *E. coli* ST95 strains was retrieved in January 2019 from the Short Read Archive (SRA) using accession numbers reported in EnteroBase[32,82]. In addition, 16 ST95 complete genomes were included in the study. To generate the 76 ST *E. coli*

database, we retrieved 100 randomly selected genomes from each ST that contained >100 genomes on Enterobase. STs from *Shigella* spp. and *Salmonella* spp. that contained >100 genomes were similarly downloaded randomly from Enterobase[32]. The final database comprised 7600 *E. coli* draft genomes from 76 STs, 1,000 *Shigella* spp. draft genomes from 10 STs, and 10,000 *Salmonella* spp. draft genomes from 100 STs. Supporting metadata from *Shigella* species and *Salmonella* serovars was retrieved from Enterobase[32].

### Quality control, mapping, variant detection and de novo assembly

The quality of Illumina reads was assessed using FastQC v0.11.6 (https://www.bioinformatics.babraham.ac.uk/projects/fastqc/) and MultiQC v1.6 (https://multiqc.info/), and reads were trimmed using Trimmomatic v0.36[83], with a minimum quality score of 10 and a minimum read length of 50. Trimmed reads were mapped to MS7163[23] using Bowtie 2 v2.3.4.2[84] and were de novo assembled using SPAdes v3.12.0[85] with default parameters. To retain only high-quality genomes, we applied the quality control as reported previously[13] and discarded genomes with >17,654 uncalled bases (mapping), >780 contigs, and genomes with a size outside the 4,811,596–5,585,756 bp range. This left 596 genomes in the final filtered dataset.

### Recombination detection and phylogenetic reconstruction

Illumina reads from the 596 strains, as well as simulated paired end reads from 16 complete genomes (1 million of 100-bp paired end reads with 500 bp−insert size, standard deviation of 50, without indels, mutations/variants and error rate), were mapped to MS7163[23] using Bowtie 2 v2.3.4.2[84]. Nesoni v0.132 (https://github.com/Victorian-Bioinformatics-Consortium/nesoni) was used to generate a core-SNP alignment for all strains with a consensus cut-off and majority cut-off of 0.90 and 0.70, respectively.

Regions of recombination were detected using Gubbins v2.3.1[86]. Using previously described methods[13,39], we generated a pseudogenome for each strain by integrating its predicted SNPs into the reference genome of MS7163, and used the alignment of these pseudogenomes as an input for Gubbins under the general time-reversible (GTR) GAMMA model of among-site rate variation (ASRV) with FastTree as the tree builder and 20 iterations. Predicted recombinant SNPs were removed from the core-SNP alignment, from which the Maximum likelihood phylogenetic tree was built using IQ-TREE[87] using the built-in optimal model detection[88]. The robustness of the tree was validated with 1000 bootstraps[89]. The ST95 phylogenetic tree was rooted using ED1a (accession number: CU928162) as the outgroup. Phylogenetic trees of *E. coli* strains with different STs were constructed according to their corresponding STs using parsnp v1.5.3[90]. Population structure was analysed with fastbaps[25] and popPUNK[26] with default parameters.

### Pangenome analysis, virulence factors, AMR genes and plasmid analysis

De novo assemblies of the 596 ST95 isolates and 16 complete genomes were annotated using Prokka v1.13[91] using its built-in *E. coli* database. Pangenome analysis of ST95 was performed using Roary[92] with Prokka annotated assemblies and complete genomes with the minimum percentage identity set at 95%; core genes were defined as genes present in at least 99% of the dataset. O-H serotyping, virulence factors, antibiotic resistance genes and plasmid Inc typing were performed using ABRicate (https://github.com/tseemann/abricate) with the pre-installed databases: EcOH[93], Virulence factor database VFDB[94], NCBI antimicrobial resistance genes[95], and PlasmidFinder[96], with the percentage nucleotide identity and the coverage cut-off set at 90% and 80%, respectively. In addition, chromosomal point mutations associated with antibiotic resistance were detected using PointFinder[97]. An in-house *E. coli* capsule database[98] was used to identify the capsule type of ST95 isolates employing Kaptive[99].

### In silico detection of cellulose disruption mutations

Individual genes from the *bcsRQABZC-bcsEFG* cellulose biosynthesis and pEtN modification operons and their corresponding 100-bp flanking regions were extracted from the database of draft assemblies retrieved from Enterobase[32] using the cellulose operon from UTI89 as a reference with a cut-off of 95% gene coverage. Open-reading frames were detected using the tool "getorf" from EMBOSS (http://emboss.open-bio.org/rel/rel6/apps/getorf.html). Genes were defined as truncated if the length of the detected ORF was <99% of the length of full protein. Precise mutations were determined by aligning the nucleotide sequence of each respective cellulose gene with their corresponding gene in UTI89.

### Bacterial strains, growth conditions and targeted gene mutagenesis

MS7163 was isolated from a patient with severe pyelonephritis and belongs to the O45:K1:H7 serotype[23]. Isogenic MS7163 derivative strains are listed in Supplementary Data 6. Strains were grown at 37 °C on solid or in liquid Lysogeny broth (LB) medium unless otherwise indicated. Chloramphenicol (30 μg/ml) or kanamycin (50 μg/ml) were added as required. In assays that assessed the contribution of curli and cellulose, strains were grown in LB without salt or on YESCA agar to induce for curli and cellulose production.

### Correction of BcsA frameshift mutation

Genome editing to correct the cellulose mutation in MS7163 was performed using the pORTMAGE system as previously described[100,101], with primers listed in Supplementary Data 6. A colony that possessed a cellulose-positive phenotype was identified and referred to as *bcsA\**; whole genome sequencing confirmed correction of the *bcsA* mutation and the absence of additional off-target mutations.

### K1 ELISA

K1 capsule expression was detected by ELISA using an anti-polysialic acid antibody single chain Fv fragment[29] as the primary antibody (1:100 dilution), anti-His antibody (ThermoFisher Scientific, 1:1,000 dilution) and alkaline phosphatase anti-mouse IgG (Alexa Fluor™ 647 #A28181, ThermoFisher Scientific, 1:10,000 dilution) as the secondary and tertiary antibodies, respectively; p-nitrophenylphosphate (Sigma) was used as the substrate. Optical density was measured at 405 nm.

### Neonatal rat disseminated infection model

All experiments were performed using the following housing conditions: (light:dark cycle 12:12 h, room temperature 21 ± 1 °C, humidity 50 ± 10%). One-day old Wistar rat (sex not reported) were infected by oral gavage with $10^7$ CFU of MS7163 (WT), MS7163*bcsA\** (*bcsA\**), or PBS (control). Litter sizes ranged from 8–16 pups, with all pups in a single litter infected with the same strain. Infected newborn rats were monitored for 7 days employing an infection severity scoresheet as previously described[102]. Pups that displayed signs of suffering above the defined severity threshold were euthanised to minimise suffering. On day 7 post-infection, remaining pups were euthanised and tissues corresponding to small intestine, caecum, large intestine, liver, spleen and brain were collected aseptically for CFU enumeration, histology and/or immunofluorescence imaging.

### Mouse model of UTI

All experiments were performed using the following housing conditions: (light:dark cycle 12:12 h, room temperature 21 ± 1 °C, humidity 50 ± 10%). The C57BL/6 female mouse model of UTI was employed as previously described[38,103]. All strains were enriched for type 1 fimbriae expression by three successive rounds of static growth in LB without salt for 48 h followed by one round of static growth for 24 h for inoculum preparation. Bacteria (~5 × 10^8 CFU in 20 μl PBS) were injected transurethrally into female C57BL/6 mice (8–10 weeks). Urine was

collected from mice every 24-h. Mice were euthanized by cervical dislocation after 24 h or 72 h. Bacterial loads corresponding to each strain in the urine, bladder and kidney were enumerated by plating onto LB agar. Statistical analyses were performed using the two-tailed Wilcoxon matched-pairs signed-rank test (Prism9, GraphPad).

## Bioplex

Mice were infected as described above and were euthanised 24-h post infection. The bladders were collected and homogenised in the presence of a protease inhibitor cocktail (Roche). The samples were spun at 12,000 x $g$ for 20 min at 4 °C, and supernatants were stored at −80 °C. Cytokine measurements were performed on 4–9 samples per group using the Bio-Plex Pro Mouse Cytokine 23–Plex Group 1 kit as per manufacturer's instructions (Bio-Rad).

## Histopathology

Tissues were fixed in 10% buffered formalin for 24 h at room temperature followed by routine processing and embedded in paraffin. Tissues were sectioned at 5 µm on a Leica microtome and stained with hematoxylin and eosin (H&E), neutrophil-specific Ly6G (Abcam, rat anti-mouse Ly-6G clone NMP-R14, 1:120 dilution) or used for immunohistochemistry. H&E and Alcian blue-periodic acid Schiff's (PAS-AB, for mucin and goblet cells) stained slides were scanned and images were captured using a VS120 Olympus digital microscope. PAS-AB stained images were analysed for goblet cell specific mucin production by ImageJ software (version 1.53).

## Immunofluorescence and imaging

Paraffin embedded tissue sections were de-waxed with a double 5-min washes in Xylene followed by rehydration with 5-min each 100% ethanol, 95% ethanol, 70% ethanol, and distilled water washes. Slides were microwaved at high temperature in sodium citrate buffer solution (10 mM citric acid and 0.05% Tween-20, pH 6.0) for 30 min and then allowed to cool for antigen retrieval. After PBS washes, samples were incubated with PBS-T (PBS with 0.01% Tween-20) containing 10% KPL blocker (milk diluent/blocking solution concentrate, Seracare) to block nonspecific binding sites followed by incubation with mouse-E-cadherin antibody (Abcam, ab231303) or rabbit anti-Iba1 antibody (NovaChem, 019-19741) (both 1:500 dilution in 5% KPL buffer) in a humidified chamber at 4 °C overnight. Samples were then rinsed with PBS-T, incubated with anti-rabbit AF647 (#A-21244) or AF488 (#A-11008) secondary antibodies for 1 hr at room temperature (1:500 5% KPL buffer, ThermoFisher Scientific) and counter stained with DAPI (Sigma-Aldrich, 1:1000) for 15 min. The slides were mounted with cover slips and analysed under confocal laser scanning microscope (Olympus FV3000). Confocal images were analysed for relative quantification of fluorescent markers by ImageJ software (version 1.53).

## Brain tissue preparation for histology and immunostaining

Paraffin-embedded brains were sectioned (16 µm) using a rotary microtome (Leica). Paraffin-fixed serial brain sections were mounted and dried on slides for de-waxing. De-waxed brain sections were washed with PBS and fixed using 4% paraformaldehyde for 5 min. The sections were then blocked in 5% (v/v) normal goat serum (NGS) with 0.3% (v/v) Triton X-100 (Sigma-Aldrich) in PBS, before being incubated overnight in a humidified chamber at 4 °C with rabbit anti-IBA1 (1:750; NovaChem). The following day, sections were washed three times in PBS for 10 min and incubated in 3% (v/v) NGS with 0.1% (v/v) Triton X-100 in PBS with goat anti-rabbit Alexa Fluor 647 (1:1000; Thermo-Fisher Scientific) for 2 hrs. Vectashield H-100 Mounting Medium (Vector Laboratories) was applied and sealed by placing a glass coverslip over sections (Menzel Glaser) and the edges of the coverslip were coated with clear nail polish.

## Brain imaging and cell quantification

For the confocal imaging of Iba1[pos] microglial cells in the parenchyma and the pia layer of the meninges, three consecutive sagittal sections of the whole brain were imaged on a spinning-disk confocal microscope (Innovative Instruments Inc.; with W1 spinning-disk module, Yokogawa), using a 20× 1.2 NA air-objective, a Hamamatsu Flash 4.0 sCMOS camera, and running SlideBook software (version 6.0.16; Innovative Instruments Inc.). For quantification, a 16-bit 3D montage of the full 16 µm physical depth of the section was acquired with a z-interval of 0.4 µm and compressed into a maximum z-projection. For all imaging of Iba1[pos] cells in the cortex or pia mater layer, the cell counts, mean fluorescence intensities, and cell morphologies were quantified using ImageJ (version 1.53, Fiji) software by selecting a region with the same 20X magnification. Cells were counted using the Cell Counter plugin, which were then normalized to the 3D area of the ROI, and the mean cell counts were averaged for each animal. For cells in the pia mater, counts were normalized to the length of the ROI. Analysis of Iba1 fluorescence intensity was conducted by calculating the corrected total cell fluorescence from five cells, normalising the intensity by subtracting all background signal to allow uniform comparison of Iba1 fluorescence intensity between samples. Iba1[pos] microglia morphology in the brain parenchyma was analysed using the skeletonize plugin. Iba1[pos] cells were selected and skeletonized to extrapolate quantitative data of endpoints and process length from the soma of each microglia.

## In-situ cell death assay (TUNEL)

Apoptosis was assessed in brain, liver, SI and colon tissues using the terminal deoxynucleotide transferase-mediated dUTP nick-end labelling (TUNEL) assay following manufacturer's instructions (Click-iT™ TUNEL Assay Kit, Invitrogen). Samples were analysed under a confocal laser scanning microscope (Olympus FV3000) after successful staining with TUNEL assay kit. Images were analysed for relative quantification of apoptosis fluorescent marker by ImageJ software (version 1.53).

## Isolation of murine intestinal lamina propria lymphocytes

Intestines were collected and cut open longitudinally and intestinal luminal contents were washed with cold PBS, then snipped into roughly 1 cm pieces. The pieces were stirred at 37 °C for 20 min in HBSS supplemented with 10% FBS, penicillin/streptomycin and 3 mM EDTA to wash off mucus and epithelial layer. Residual tissue was cut into fine pieces and stirred in RPMI 1640 supplemented with 10% FBS, Collagenase D (Roche) and Dnase I (Sigma-Aldrich) at 37 °C for 40 min. The tissue suspension was passed through a 100 µm cell strainer (Corning) and washed with RPMI before layering on a 40% Percoll (Sigma-Aldrich) density gradient solution. The lamina propria lymphocytes (LPLs) in the pellet were collected and washed with RPMI 1640 and stained for flow cytometry.

## Flow cytometry

The brain, liver and intestinal cell suspensions were washed with protein-free saline before staining with live/dead viability FVS700 (BD Horizon) at 4 °C for 15 min. Cells were then washed with ice cold MACS Buffer (Miltenyi Biotech), and stained for surface antigens for 20 min at 4 °C. The following fluorochrome-conjugated anti-rat monoclonal antibodies were used at 1:1000 dilution: CD45 PE Cy7 #561588 (OX-1; BD Biosciences), CD4 APC Cy7 #201518 (W3/25; Biolegend), CD45R BV785 #743595 (HIS24; BD Biosciences), CD43 APC #202810 (W3/13; Biolegend), CD161a BUV395 #744055 (8764; BD Biosciences), CD172A BV421 #744861 (OX-41; BD Biosciences), CD11b/c BV510 #743978 (OX-42; BD Biosciences), His-48 FITC #554907 (HIS48; BD Biosciences). For intracellular staining, cells were fixed and permeabilized using eBioscience™ Foxp3/Transcription Factor Staining Buffer set followed by incubation with monoclonal antibodies for 30 min at 4 °C. Samples

were acquired on an LSRII Fortessa flow cytometer (BD Biosciences) and analysed with FlowJo software. Gating strategies for flow cytometric analyses of live immune cells are provided in Supplementary Figs. 9, 10.

## Statistics and reproducibility
Statistical analyses were done using One-Way ANOVA, Mann-Whitney or Chi-squared test as part of GraphPad Prism 9. No statistical method was used to predetermine sample size. No data were excluded from the analyses. Groups of mice/rats were determined at random. The investigators were not blinded to allocation during experiments and outcome assessment.

## Data availability
The accession number of ST95 genomes is included in Supplementary Data 1. The accession number for the genomes representing the 76 *E. coli* STs, 10 Shigella STs and 100 Salmonella STs randomly downloaded from Enterobase are provided in Supplementary Data 7. Source data are provided with this paper.

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

## Acknowledgements

We thank Thom Cuddihy, Rhys White and Barb Arnts for technical support. This work was supported by Australian National Health and Medical Research Council (NHMRC) Ideas grants 1181958 and 2001431 (to M.A.S., M.D.P. and N.T.K.N.), 2013776 (to J.V.), and 2021475 (to M.J.Su., K.G.K.G., and G.C.U.), as well as grants from the Gastroenterological Society of Australia (to S.Z.H.), the Australian Infectious Diseases Research Centre (to S.Z.H., and to M.A.S., A.D.I. and N.T.K.N.), The University of Queensland (to MAS and S.Z.H.), and the Mater Foundation (to S.Z.H.). J.V. is supported by a Senior Research Fellowship from the Sylvia and Charles Viertel Foundation. S.Z.H. was supported by an NHMRC Fellowship (2018–2022) and a University of Queensland Amplify Fellowship (2022–2024). M.J.Sw is supported by an NHMRC Investigator grant (APP1194406).

## Author contributions

Conceptualisation: N.T.K.N., J.V, G.C.U., S.Z.H., M.A.S. Investigation: N.T.K.N., M.A.R., K.G.K.G., S.J.K., M.-D.P., K.M.P., L-A.F., S.J.H., C.R., T.J.K. and M.J.Su.. Formal analysis: N.T.K.N., M.A.R., K.G.K.G., S.J.K., M.J.Sw, A.D.I., J.V., G.C.U., S.Z.H., M.A.S. Supervision: K.M.I., S.A.B., M.J.Sw., A.D.I., J.V., G.C.U., S.Z.H., M.A.S. Writing—original draft: N.T.K.N., M.A.R., K.G.K.G., S.J.K., J.V., G.C.U., S.Z.H., M.A.S.. Writing—review and editing: all authors. All authors read and approved the final manuscript.

## Competing interests

The authors declare no competing interests.

## Additional information

[1]Institute for Molecular Bioscience (IMB), The University of Queensland, Brisbane, QLD, Australia. [2]School of Chemistry and Molecular Biosciences, The University of Queensland, Brisbane, QLD, Australia. [3]Australian Infectious Diseases Research Centre, The University of Queensland, Brisbane, QLD, Australia. [4]Immunopathology Group, Mater Research Institute, The University of Queensland, Translational Research Institute, Brisbane, Australia. [5]School of Pharmacy and Medical Sciences, Griffith University, Southport, QLD, Australia. [6]Menzies Health Institute Queensland, Griffith University, Southport, QLD, Australia. [7]School of Biomedical Sciences, Faculty of Medicine, The University of Queensland, Brisbane, QLD, Australia. [8]Queensland Brain Institute, The University of Queensland, Brisbane, QLD, Australia. [9]University of Queensland Centre for Clinical Research, Brisbane, Australia. [10]Queensland Children's Hospital, Brisbane, Australia. [11]Present address: QIMR Berghofer Medical Research Institute, Brisbane QLD, Australia. [12]Present address: INRAE, Univ Montpellier, LBE, 102 Avenue des Etangs, Narbonne 11100, France. [13]Present address: Wellcome-Wolfson Institute for Experimental Medicine, School of Medicine, Dentistry and Biomedical Sciences, Queen's University Belfast, Belfast, UK. [14]Present address: Central Microbiology, Pathology Queensland, Royal Brisbane and Women's Hospital, Brisbane, Australia. [15]Present address: School of Biological Sciences, University of East Anglia, Norwich NR4 7TJ, UK. [16]These authors contributed equally: Nguyen Thi Khanh Nhu, M. Arifur Rahman, Kelvin G. K. Goh, Seung Jae Kim. ✉e-mail: j.vukovic@uq.edu.au; g.ulett@griffith.edu.au; sumaira.hasnain@mater.uq.edu.au; m.schembri@uq.edu.au

