## [Peer Review File · Nature Communications]

A convergent evolutionary pathway attenuating cellulose production drives enhanced virulence of some bacteriaEditorial Note: This manuscript has been previously reviewed at another journal that is not operating a transparent peer review scheme. This document only contains reviewer comments and rebuttal letters for versions considered at *Nature Communications*.

REVIEWERS' COMMENTS

Reviewer #1 (Remarks to the Author):

The authors have thoughtfully addressed all of my concerns, and I recognize the work that was put in to the revision.

Reviewer #2 (Remarks to the Author):

This is the revised version of a manuscript previously submitted to Nature Communications. The manuscript has greatly improved however, I still have some major comment. The authors provide the impression that the promotion of virulence by disruption of cellulose biosynthesis in Enterobacteriaceae is an original finding. This is not the case. At several instances, mutational analysis has shown that modulation of cellulose biosynthesis in Salmonella significantly alters the degree of virulence. Those studies need to be included in the discussion. Pontes MH, Lee EJ, Choi J, Groisman EA. Salmonella promotes virulence by repressing cellulose production. Proc Natl Acad Sci U S A. 2015 Apr 21;112(16):5183-8. Ahmad I, Rouf SF, Sun L, Cimdins A, Shafeeq S, Le Guyon S, Schottkowski M, Rhen M, Römling U. BcsZ inhibits biofilm phenotypes and promotes virulence by blocking cellulose production in Salmonella enterica serovar Typhimurium. Microb Cell Fact. 2016 Oct 19;15(1):177. Petersen E, Mills E, Miller SI. Cyclic-di-GMP regulation promotes survival of a slow-replicating subpopulation of intracellular Salmonella Typhimurium. Proc Natl Acad Sci U S A. 2019 Mar 26;116(13):6335-6340.

Reviewer #3 (Remarks to the Author):

I previously reviewed this article (reviewer 3) and in the interests of fairness I have reviewed this revised version solely in the light of my previous comments and not those of the other reviewers.

I thank the authors for their consideration of my previous points. The PopPunk analysis adds statistical rigour to what was previously arbitrary eyeball clustering. However my original point remains. What is presented is an account of a mutation in a single clade of ST95 and how it ablates cellulose production, and then correlative assumptions that other mutations do the same thing. I also still believe the title is misleading and needs softening, stating you have kept it to engage interest is almost saying as much as agreeing with my point

RESPONSE TO REVIEWER COMMENTS

Reviewer #1 (Remarks to the Author):

The authors have thoughtfully addressed all of my concerns, and I recognize the work that was put in to the revision.

We thank the reviewer for positive comments on our manuscript

Reviewer #2 (Remarks to the Author):

This is the revised version of a manuscript previously submitted to Nature Communications. The manuscript has greatly improved however, I still have some major comment.

The authors provide the impression that the promotion of virulence by disruption of cellulose biosynthesis in Enterobacteriaceae is an original finding. This is not the case. At several instances, mutational analysis has shown that modulation of cellulose biosynthesis in Salmonella significantly alters the degree of virulence. Those studies need to be included in the discussion.

Pontes MH, Lee EJ, Choi J, Groisman EA. Salmonella promotes virulence by repressing cellulose production. Proc Natl Acad Sci U S A. 2015 Apr 21;112(16):5183-8.

Ahmad I, Rouf SF, Sun L, Cimdins A, Shafeeq S, Le Guyon S, Schottkowski M, Rhen M, Römling U. BcsZ inhibits biofilm phenotypes and promotes virulence by blocking cellulose production in Salmonella enterica serovar Typhimurium. Microb Cell Fact. 2016 Oct 19;15(1):177.

Petersen E, Mills E, Miller SI. Cyclic-di-GMP regulation promotes survival of a slow-replicating subpopulation of intracellular Salmonella Typhimurium. Proc Natl Acad Sci U S A. 2019 Mar 26;116(13):6335-6340.

We thank the reviewer for the suggestion. We have tempered claims that our work describes an original finding in Salmonella and have cited the papers suggested by the reviewer in the discussion.

Lines 431-436: In Salmonella, previous studies have shown that cellulose disruption promotes virulence⁷⁹⁻⁸¹. Here, we extended these findings and demonstrate that cellulose disruption mutations occur in all human-restricted *S. enterica* serovars causing typhoid fever (*S. Typhi*) and paratyphoid fever (*S. Paratyphi* A, B and C). Interestingly, disruption of the BcsG pEtN transferase is associated with clonal replacement of *Salmonella* Typhimurium ST313 lineage 1 by lineage 2, which is a major cause of bloodstream infection in Africa^{77,78}. Thus, loss-of-function in cellulose production is a pathoadaptive mechanism that shares a correlative association with many Enterobacteriaceae causing severe human infection.

Reviewer #3 (Remarks to the Author):

I previously reviewed this article (reviewer 3) and in the interests of fairness I have reviewed this revised version solely in the light of my previous comments and not those of the other reviewers.

I thank the authors for their consideration of my previous points. The PopPunk analysis adds statistical rigour to what was previously arbitrary eyeball clustering. However my original point remains. What is presented is an account of a mutation in a single clade of ST95 and how it ablates cellulose production, and then correlative assumptions that other mutations do the same thing. I also still believe the title is misleading and needs softening, stating you have kept it to engage interest is almost saying as much as agreeing with my point.

As indicated above, we have tempered our claims by modifying several sections of the manuscript as follows (changes underlined):

Title.

A convergent evolutionary pathway attenuating cellulose production drives enhanced virulence of some bacteria.

Abstract.

Mutations that disrupt cellulose production were also identified in other virulent ExPEC STs, *Shigella* and *Salmonella*, suggesting a correlative association with many *Enterobacteriaceae* that cause severe human infection.

Discussion.

L439-440. Thus, loss-of-function in cellulose production is a pathoadaptive mechanism that shares a correlative association with many *Enterobacteriaceae* that cause severe human infection.